# Unremovable Watermarks for Open-Source Language Models

## Abstract

The recent explosion of high-quality language models has necessitated new methods for identifying AI-generated text. Watermarking is a leading solution and could prove to be an essential tool in the age of generative AI. Existing approaches embed watermarks at inference and crucially rely on the large language model (LLM) specification and parameters being secret, which makes them inapplicable to the open-source setting. In this work, we introduce the first watermarking scheme for open-source LLMs. Our scheme works by modifying the parameters of the model, but the watermark can be detected from just the outputs of the model. Perhaps surprisingly, we prove that our watermarks are *unremovable* under certain assumptions about the adversary's knowledge. To demonstrate the behavior of our construction under concrete parameter instantiations, we present experimental results with OPT-6.7B and OPT-1.3B. We demonstrate robustness to both token substitution and perturbation of the model parameters. We find that the stronger of these attacks, the model-perturbation attack, requires deteriorating the quality score to 0 out of 100 in order to bring the detection rate down to 50%.

## 1 Introduction

As generative AI becomes increasingly capable and available, reliable solutions for identifying AI-generated text grow more and more imperative. Without strong identification methods, we face risks such as model collapse (Shumailov et al., 2024), mass disinformation campaigns (Ryan-Mosley, 2023), and detection false positives leading to false plagiarism accusations (Ghaffary, 2023). Watermarking is a prominent and promising approach to detection. Recent works (Kirchenbauer et al. (2023); Aaronson (2022); Zhao et al. (2024); Christ et al. (2024); Kuditipudi et al. (2024); Fairoze et al. (2023); Christ & Gunn (2024)) construct "sampler-based watermarks," for the setting where the watermark is embedded at generation time and users have only query access to the model. Such watermarks modify the algorithm used by the LLM to sample from the probability distribution over the next token, leaving the underlying neural network untouched. For example, Zhao et al. (2024) shifts this distribution to prefer sampling tokens on a fixed "green list." These existing approaches are ill-suited for scenarios where an attacker has access to the code for the watermarked model and can simply rewrite the sampling algorithm to sample from the unmodified probability distribution — yielding the original, unwatermarked model, with no loss in quality.

Such scenarios are gaining importance as open-source models become more widely available and higher quality (e.g., LLaMA Touvron et al. (2023), Mistral Jiang et al. (2023), OPT Zhang et al. (2022)). However, watermarks for open-source models have been severely understudied. In this work we initiate a formal study of such watermarks, in the setting where the model parameters and associated code are publicly available, and the watermark is embedded by altering the weights of the neural network. We consider the standard notion of autoregressive language models used in existing watermarking works (see Appendix A.2 for a precise definition). Constructing watermarks that are unremovable by an attacker with such comprehensive model access is a significant challenge.

**Our contributions.** We put forth a formal definition of an unremovable watermark for neural networks, then construct such a watermarking scheme. Unlike sampler-based watermarks, ours modifies the weights of the model. Our watermark is unremovable in the sense that it can be detected even after an adversary modifies the weights of the model. In fact, it offers unremovability not only with access to the weights, but even from a modest amount of text (∼300 tokens) produced by the model.

Of course, one must place a restriction on the allowable ways in which the adversary may modify the weights, otherwise it can ignore the watermarked model and generate unwatermarked text of its own. We do so by imposing a quality condition: We prove that if an adversary removes our watermark, the resulting model is *low-quality*, assuming that the attacker has some uncertainty about the distribution of high-quality text. This assumption is necessary because otherwise, the adversary could simply ignore the watermarked model and train a new model itself. We precisely formalize this assumption and the tradeoff between quality degradation and watermark strength.

In addition to proving unremovability, we provide a suite of experiments using OPT-6.7B and OPT-1.3B, showing that our theoretical guarantees translate to practice. We find that the strongest attack we consider requires deteriorating text quality to zero out of 100 in order to bring the detection rate to 50%. Our approach is quite general, and our techniques may be of independent interest due to their applicability in other settings where one wishes to watermark high-dimensional data.

## 2 TECHNICAL OVERVIEW

**Formalizing unremovability (Section 4.1).** The first challenge in constructing an unremovable watermark is defining unremovability. Recall that the goal of the adversary is to take a watermarked model and produce a high-quality unwatermarked model. One cannot hope for a watermark to be unremovable by an adversary with sufficient knowledge to train its own unwatermarked model. Therefore, we must consider only adversaries *with limited knowledge about the distribution of high-quality text*. We formalize this notion as follows: we model ideal-quality language as being described by an original neural network $M^*$. That is, the ideal-quality language is the distribution of text that would be produced by an LLM using $M^*$ to compute the probability distribution over each token. This neural network $M^*$ is then watermarked, and the resulting $M'$ is given to the adversary. The adversary's uncertainty is captured by its *posterior distribution over $M^*$*, given $M'$.

We assume that the adversary's posterior distribution over $M^*$ is normally distributed in a particular representation space of models, and centered at $M'$. Under this assumption, we prove that there is a steep trade-off between the magnitude of changes the adversary must make to remove the watermark, and the magnitude of changes required to add the watermark. That is, text produced by the adversary with access to $M'$ is either watermarked or low-quality.

**Our scheme (Section 5).** Our scheme is quite natural: We modify the bias of each neuron in the last layer of the model by adding a perturbation from $\mathcal{N}(0, \varepsilon^2)$, as described in Algorithms 1 and 2. The watermarking detection key is the vector of these perturbations. Given the weights of a model, we can detect the watermark by checking the correlation between the biases of its last layer and the watermark perturbations. More precisely, we compute the inner product between the watermark perturbations, and the difference between the biases of the given model and the original model (Algorithm 3).

Furthermore, we observe that it is possible to approximate this inner product *given only text* produced by the model. This text detector (Algorithm 4) computes a score that is the sum of the watermark perturbations corresponding to the distinct tokens observed in the text. We show that in expectation, this score is approximately the inner product from the detector from weights (Algorithm 3), where the approximation error is lower for higher-entropy text. We prove that in sufficiently high-entropy text, our text detector succeeds with overwhelming probability.

**Proving unremovability (Theorem 3).** We show unremovability in two steps: (1) Any high-quality model produced by altering the watermarked model must have a high inner product score (Theorem 1), and (2) If a language model has a high inner product score, its responses are watermarked (Theorem 3).

To understand the intuition behind (1), consider the watermarked and unwatermarked models as vectors $\vec{w}_{\mathsf{wat}}, \vec{w}^* \in \mathbb{R}^n$, respectively. The adversary is given $\vec{w}_{\mathsf{wat}}$, and it aims to produce $z \in \mathbb{R}^n$ that is unwatermarked, but that is sufficenty close to $\vec{w}^*$ (i.e., has high quality). We can describe the region of watermarked vectors geometrically: This region is essentially a halfspace, where the hyperplane describing it lies between $\vec{w}_{\mathsf{wat}}$ and $\vec{w}^*$, and it is orthogonal to the line from $\vec{w}_{\mathsf{wat}}$ to $\vec{w}^*$. Removing the watermark involves crossing this hyperplane. If $\vec{w}^*$ is known, the most efficient way to remove the watermark is to add to $\vec{w}^*$ in the direction of $(\vec{w}^* - \vec{w}_{\mathsf{wat}})$. However, because the adversary's posterior distribution over $\vec{w}^*$ is centered at $\vec{w}_{\mathsf{wat}}$, the adversary is unlikely to produce

a short vector $(z - \vec{w}_{\text{wat}})$ that moves far in the direction of $(\vec{w}^* - \vec{w}_{\text{wat}})$: We expect its weight in this direction to be proportional to its length. This allows us to show that, if $z$ is not watermarked, then $(z - \vec{w}_{\text{wat}})$ has large magnitude. If $(z - \vec{w}_{\text{wat}})$ has large magnitude, then $z$ is far from $\vec{w}^*$ and therefore low quality.

Now consider some text produced by a model on the watermarked side of the hyperplane. For each token, if the watermark increases its bias relative to that of $\vec{w}^*$, the model is more likely to output that token. Prior work of Zhao et al. (2024) uses this principle to obtain a simple "counting" detector, which essentially computes the fraction of positive-bias tokens in the given response. However, we observe that because our watermark alters different token biases by different amounts, we can substantially improve detectability by taking the magnitudes of the perturbations into account in our detector. Our detector computes a score, which is the sum of bias perturbations of all observed tokens in the text (in contrast to the counting detector, which is the sum of indicators of the bias perturbations' *signs*). Interestingly, we prove that our text detector, when applied to text produced by $z$, approximates the inner product between $(z - \vec{w}^*)$ and $(\vec{w}_{\text{wat}} - \vec{w}^*)$. Because this inner product is high for watermarked $z$ and roughly 0 for independently generated $z$, our detector is reliable and has a negligible false positive rate. Similarly to other watermarks (e.g., Kirchenbauer et al. (2023); Christ et al. (2024); Zhao et al. (2024)) our proof of detectability from text (2) relies on sufficient entropy in the text.

When proving unremovability of the text watermark (2), we assume that the adversary alters only the biases in the last layer of the watermarked model it is given. We argue that an adversary successfully making arbitrary alterations to the model could use this new model to learn ways to alter only the biases of the watermarked model. We leave a formal expansion of the class of adversaries to future work.

**Experiments (Section 6).** We show that our watermark is indeed detectable and unremovable in practice, using experiments run on OPT-6.7B and OPT-1.3B (Zhang et al., 2022). We use Mistral-7B Instruct (Jiang et al., 2023) to measure the quality of responses, and we show that the watermarked models with our scheme preserve similar quality to their unwatermarked version. We demonstrate reliable detectability rates for our construction, given $\sim$300-token responses of various types. In addition to our proofs for unremovability, we consider a number of concrete adversaries and demonstrate that the watermark persists in the adversarially altered watermarked models unless the adversary adds large perturbations that significantly reduce the quality of the resulting model.

## 3 RELATED WORK

We describe the most relevant related works here, but refer readers to Tang et al. (2023); Piet et al. (2023) for more comprehensive surveys.

Existing watermarks change the sampling function of the LLM, and are therefore easily removable given code for this sampling function. Aaronson (2022); Kirchenbauer et al. (2023); Zhao et al. (2024); Christ et al. (2024); Fairoze et al. (2023) sample each token by partitioning the token set into red and green lists (that depends on the previously-output tokens), then increasing the probability of the tokens in the green list. To detect the watermark, one computes the fraction of green tokens in the given text and compares it to a threshold. Kuditipudi et al. (2024); Christ & Gunn (2024) operate slightly differently—for each response, they choose a random seed. In Kuditipudi et al. (2024) there are polynomially many possible seeds for a given watermarking key, and in Christ & Gunn (2024) there are exponentially many. They then choose the $i^{\text{th}}$ token of the response to be correlated with the $i^{\text{th}}$ value of the random seed. To detect the watermark, one essentially computes the correlation between the response and the seed. While not unremovable, some of these watermarks satisfy a weaker *robustness* property: Given a watermarked response, it is difficult to make a bounded number of edits to yield an unwatermarked text.

Our scheme is most similar in spirit to that of Zhao et al. (2024), though theirs is a sampler-based watermark. They choose a fixed red/green partition that is used to change the sampling function for all responses. Similarly, we fix the partition of tokens whose biases we increase and decrease. The two major differences are that (1) we make this change by altering the weights of the model, and (2) while Zhao et al. (2024) increases the logits of all green tokens by the same amount, our increases are normally distributed. This normal distribution is crucial for unremovability, and it results in a different detector being most effective for our scheme. The detector of Zhao et al. (2024) simply

computes the fraction of green tokens, while our inner product detector computes a score that is essentially a weighted count of the number of green tokens. These weights correspond to the size of the perturbations added by our watermark.

The works closest to our setting are Gu et al. (2024); Sander et al. (2024), which show empirically that certain sampler based watermarks can be learned. That is, a model trained on watermarked texts may learn to generate watermarked responses itself. The watermark therefore is embedded in the weights of the model; however, there is no unremovability guarantee, and the detectability of these responses is much lower than their sampler-based counterparts. In fact, Gu et al. (2024) finds empirically that the watermark is destroyed after the model is fine-tuned and leaves the question of unremovable open-source watermarks for future work.

We also note that prior work Zhang et al. (2024) shows that any LLM watermark is removable by a sufficiently strong adversary. This underscores our need to consider an adversary with limited knowledge about high-quality language when defining and showing unremovability.

## 4 PRELIMINARIES AND DEFINITIONS

Let $\mathbb{N} := \{1, 2, \dots\}$ denote the set of positive integers. We will write $[q] := \{1, \dots, q\}$. For a set $X$, we define $X^* := \{(x_1, \dots, x_k) \mid x_1, \dots, x_k \in X \wedge k \in \mathbb{Z}_{\geq 0}\}$ to be the set of all strings with alphabet $X$. For a binary string $s \in X^*$, we let $s_i$ denote the $i^{\text{th}}$ symbol of $s$ and $\text{len} s$ denote the length of $s$. For a string $s \in X^*$ and positive integers $a \leq b \leq \text{len} s$, let $s[a : b]$ denote the substring $(s_a, \dots, s_b)$. We use $\log(x)$ to denote the logarithm base 2 of $x$, and $\ln(x)$ to denote the natural logarithm of $x$.

For a finite set $X$, we will use the notation $x \leftarrow X$ to denote a uniformly random sample $x$ from $X$. If $X$ is a set of $n$-dimensional column vectors, we will write $X^m$ to refer to the set of $n \times m$ matrices whose columns take values in $X$. Unless otherwise specified, vectors are assumed to be column vectors. For a vector $x \in \mathbb{R}^n$, we let $\|x\| = \|x\|_2 = \sqrt{\sum_{i=1}^{n} x_i^2}$.

Let $\text{Ber}(p)$ be the Bernoulli distribution on $\{0, 1\}$ with expectation $p$. Let $\text{Ber}(n, p)$ be the distribution on $n$-bit strings where each bit is an i.i.d sample from $\text{Ber}(p)$. For a distribution $\mathcal{D}$, we let $\text{Supp}(\mathcal{D})$ denote its support.

Let $\lambda$ denote the security parameter. A function $f$ of $\lambda$ is *negligible* if $f(\lambda) = O(\frac{1}{\text{poly}(\lambda)})$ for every polynomial $\text{poly}(\cdot)$. We write $f(\lambda) \leq \text{negl}(\lambda)$ to mean that $f$ is negligible. We say a probability is *overwhelming* in $\lambda$ if it is equal to $1 - f$ for some negligible function $f$. We let $\approx$ denote computational indistinguishability and $\equiv$ denote statistical indistinguishability.

We provide a formal description of a language model in Appendix A.2.

### 4.1 WATERMARKS

We present general definitions for watermarking content from some content distribution $\mathcal{C}$ of real vectors; that is, $\text{Supp}(\mathcal{C}) \subseteq \mathbb{R}^n$. We will later use this real-content watermark framework to watermark the weights of a neural network.

**Definition 1** (Watermark). A watermark is a tuple of polynomial-time algorithms $\mathcal{W} = (\text{Setup}, \text{Watermark}, \text{Detect})$ such that:

- $\text{Setup}(1^\lambda) \to \text{sk}$ outputs a secret key, with respect to a security parameter $\lambda$.

- $\text{Watermark}_{\text{sk}}(x) \to x'$ is a randomized algorithm that takes as input some content $x$ and outputs some watermarked content $x'$.

- $\text{Detect}_{\text{sk}}(x') \to \{\text{true}, \text{false}\}$ is an algorithm that takes as input some content $x'$ and outputs true or false.

**Definition 2** (Quality loss function). A *quality loss function* for content from $\mathbb{R}^n$ is a function $L : \mathbb{R}^n \times \mathbb{R}^n \to \mathbb{R}$. We say that $L(x, y)$ is the *quality loss score* of the content $y \in \mathbb{R}^n$ relative to some ideal content $x$.

The following removability game is defined over a content distribution $\mathcal{C}$, a quality measure $L$ over $\mathbb{R}^n$, and a loss parameter $\ell(\cdot) : \mathbb{Z} \to \mathbb{R}$.

**Definition 3** (($\mathcal{C}, L, \ell$)-Removability game $\mathcal{G}^{\text{remov}}_{\mathcal{A},\mathcal{W},\mathcal{C}}(1^\lambda, L, \ell)$)**.** Let $\mathcal{C}$ be a content distribution. The removability game $\mathcal{G}^{\text{remov}}_{\mathcal{A},\mathcal{W},\mathcal{C}}(1^\lambda, L, \ell)$ is defined between an adversary $\mathcal{A}$ and a challenger as follows.

1. The challenger runs $\text{sk} \leftarrow \text{Setup}(1^\lambda)$.

2. The challenger chooses some original content $x \leftarrow \mathcal{C}$ and produces watermarked content $x_{\text{wat}} \leftarrow \text{Watermark}_{\text{sk}}(x)$.

3. $\mathcal{A}$ receives $x_{\text{wat}}$ and produces $x'$.

4. The adversary wins if $x'$ is not watermarked, and its quality loss is acceptable: $L(x, x') \leq \ell(n)$.

The output of the game $\mathcal{G}^{\text{remov}}_{\mathcal{A},\mathcal{W},\mathcal{C}}(1^\lambda, \mathcal{C}, L, \ell)$ is 1 if and only if the adversary wins.

**Definition 4** (Unremovability)**.** Let $\mathcal{W}$ be a watermark for content in $\mathbb{R}^n$. Let $\mathcal{C}$ be a content distribution, $Q$ be a quality loss function over $\mathbb{R}^n$, and $\ell(\cdot) : \mathbb{R}^n \times \mathbb{R}^n \to \mathbb{R}$ be a loss parameter. We say that $\mathcal{W}$ is ($\mathcal{C}, L, \ell$)-*unremovable* if for all p.p.t. adversaries $\mathcal{A}$,

$$\Pr\left[\mathcal{G}^{\text{remov}}_{\mathcal{A},\mathcal{W},\mathcal{C}}(1^\lambda, L, \ell) = 1\right] \leq \text{negl}(\lambda).$$

**Definition 5** (Soundness/low false positive rate)**.** Let $\mathcal{W}$ be a watermark for content in $\mathbb{R}^n$. $\mathcal{W}$ is *sound* if for any fixed $x \in \mathbb{R}^n$,

$$\Pr_{\text{sk} \leftarrow \text{Setup}(1^\lambda)}\left[\text{Detect}_{\text{sk}}(x) = \text{true}\right] \leq \text{negl}(\lambda).$$

As shorthand, when the setup and watermark algorithms are clear from context, we sometimes write that Detect (rather than $\mathcal{W}$) is unremovable or sound.

## 5 WATERMARKING SCHEME

We first show a general watermarking scheme for vectors in $\mathbb{R}^n$. We then show how to apply this paradigm to neural networks, by watermarking the biases of the output layer. Our watermark detector is strongest when given explicit access to the weights of the watermarked model (Section 5.1). Furthermore, for a language model, where these biases directly affect the distribution of tokens in its outputs, our watermark is even detectable in text produced by a watermarked model (Section 5.2).

### 5.1 DETECTING THE WATERMARK FROM THE WEIGHTS OF THE MODEL

We consider an arbitrary vector $\vec{w}^*$ in $\mathbb{R}^n$ to which Gaussian noise is added, to obtain $\vec{w}_{\text{wat}}$. Observe that the inner product between $(\vec{w}_{\text{wat}} - \vec{w}^*)$ and the vector of the Gaussian perturbations $\Delta$ is large. Naturally, our detector WeightDetect (Algorithm 3) computes this inner product. We show that any adversary whose posterior distribution (after seeing $\vec{w}_{\text{wat}}$) over $\vec{w}^*$ is Gaussian cannot remove the watermark without adding significant perturbations of its own. In particular, if $\vec{w}^*$ has dimension $n$, the adversary must add a vector of Euclidean norm $\Omega(n/\sqrt{\log n})$ to succeed at removing the watermark. In contrast, embedding the watermark involves adding a perturbation of Euclidean norm only $O(\sqrt{n})$: Watermark removal requires a change that is larger than watermark embedding *by a factor of nearly* $\sqrt{n}$. So far, this approach is general to watermarking any real vector. Although our work focuses on applying it to LLMs, it is likely that this paradigm can be applied to other scenarios, which we leave for future work.

To apply our watermark to LLMs, we will later take $\vec{w}^*$ to be the biases in the last layer of a neural network, making $n$ the size of the token alphabet.

**Definition 6** ($L_2$: Euclidean quality loss function.)**.** Let $L_2 : \mathbb{R}^n \times \mathbb{R}^n \to \mathbb{R}$ be the quality loss function that, on input $x, y \in \mathbb{R}^n$, outputs $\|x - y\|_2$. We call $L_2$ the *Euclidean loss function*.

**Theorem 1.** *Let $I$ be the $n \times n$ identity matrix, and let $\mathcal{C}$ be such that the adversary's posterior distribution over the original content $\vec{w}^*$ after seeing $\vec{w}_{\text{wat}}$ is $\mathcal{N}(0, \varepsilon^2 I)$.*

*Let the loss parameter be $\ell(n) := \frac{\varepsilon n}{\sqrt{\log n}}$. Then WeightDetect is ($\mathcal{C}, L_2, \ell$)-unremovable.*

**Theorem 2.** WeightDetect *is sound.*

We defer the proofs of Theorems 1 and 2 to Appendices A.3.1 and A.3.2 respectively.

**Algorithm 1:** Watermarked content generator Setup

**Result:** Watermark secret key $\Delta \in \mathbb{R}^n$

1 **return** $\Delta \leftarrow \mathcal{N}(0, \varepsilon^2 I)$;

---

**Algorithm 2:** Watermarked content generator Watermark

**Input:** Content $x \in \mathbb{R}^n$ and watermark secret key $\Delta \in \mathbb{R}^n$

**Result:** Watermarked content $x' \in \mathbb{R}^n$ and original content $x$

1 $x' \leftarrow x + \Delta$;

2 **return** $x', x$;

---

**Algorithm 3:** Watermark detector WeightDetect$_\Delta$

**Input:** Content $c \in \mathbb{R}^n$, original content $x \in \mathbb{R}^n$, and a detection key $\Delta \in \mathbb{R}^n$

**Result:** true or false depending on whether $c$ is watermarked

1 **if** $(c - x) \cdot \Delta \geq \tau \varepsilon^2 n$ *and* $\|c - (x + \Delta)\| \leq \frac{1}{2}\varepsilon^2 n$ **then**

2    | **return** true;

3 **return** false;

---

**Algorithm 4:** Watermark detector TextDetect$_\Delta$

**Input:** Text $x_1, \ldots, x_\ell \in \mathcal{T}$ and detection key $\Delta \in \mathbb{R}^n$

**Result:** true or false depending on whether the text is watermarked

1 $S \leftarrow \emptyset$;

2 count $\leftarrow 0$;

3 **for** $i \in [\ell]$ **do**

4    | **if** $x_i \notin S$ **then**

5    |   | count $\leftarrow$ count $+ \Delta(x_i)$;

6    |   | $S \leftarrow S \cup \{x_i\}$;

7    | **if** $|S| \geq \lambda$ and count $\geq |S|\varepsilon^2 \tau_{\text{text}}$ **then**

8    |   | **return** true;

9 **return** false;

Figure 1: The algorithms $\mathcal{W} = (\text{Setup}, \text{Watermark}, \text{WeightDetect}, \text{TextDetect})$.

## 5.2 DETECTING THE WATERMARK FROM TEXT

In this section, we will show that the inner product from WeightDetect can be approximated given text. Our detector from text, TextDetect (Algorithm 4), simply computes the sum of the perturbations of the biases of tokens in the given text. We observe that a positive perturbation increases the probability of outputting the given token, and the magnitude of the perturbation corresponds to the magnitude of the increase. Similarly, a negative perturbation decreases a token's likelihood. Therefore, we expect the frequency of a token in a watermarked response to have an observable correlation with the sign and magnitude of the perturbation added to its bias.

We prove in Theorem 3 that this intuition is indeed correct. In particular, this sum of perturbations of observed tokens approximates a value that is 0 in expectation for natural text, and grows linearly with the text length for watermarked text. This is true even when the watermarked model is modified by an adversary attempting to remove the watermark. The accuracy of the detector's approximation depends on the entropy of the text, and the quality of the model produced by the adversary. That is, low-entropy responses and low-quality adversarial models will have lower watermark detectability.

We first introduce some notation and recall the structure of a language model. Let $z$ be the biases of the model produced by an adversary modifying the watermarked model $\vec{w}_{\text{wat}}$. Let $p_t^i$ and $q_t^i$ be the probabilities that the models $z$ and $\vec{w}^*$, respectively, assign to token $t \in [n]$ at step $i$. Let $\ell_r^i$ be the logit of token $r$ in step $i$, under the original model $\vec{w}^*$. Therefore,

$$p_t^i = \frac{e^{\ell_t^i + z_t}}{\sum_{r \in [n]} e^{\ell_r^i + z_r}} \text{ and } q_t^i = \frac{e^{\ell_t^i + \vec{w}_t^*}}{\sum_{r \in [n]} e^{\ell_r^i + \vec{w}_r^*}}$$

Recall that language models compute their probability distribution using the softmax function. Therefore, no probabilities are actually zero but are rather extremely small values, which are practically zero. We assume there is some threshold for which probabilities above that threshold are *significant*, and probabilities below that threshold are functionally zero—in practice, those tokens

are not sampled. We treat these negligible probabilities as zero. We will use the following assumptions:

**Assumption 1** (Similar normalization). $\sum_{r \in [n]} e^{\ell_r^i + z_r} \approx \sum_{r \in [n]} e^{\ell_r^i + \vec{w}_r^*}$.

**Assumption 2** (Derived from similar normalization, with concrete approximation error). There exists a very small constant $\eta$ such that for every $t \in [n]$,

$$p_t^i / q_t^i - 1 - \eta \leq (z_t - \vec{w}_t^*) \leq p_t^i / q_t^i - 1 + \eta.$$

Assumption 1 states that the under the adversary's model and the original model, the normalization factors used in the softmax function are roughly equal. Intuitively, for an attack that significantly changes this normalization factor, there is a similar attack that makes smaller renormalized changes. For example, if the adversary increases all biases by a large amount, it might as well change them by a smaller amount with the same ratios between them. Furthermore, the quality requirement bounds the adversary's changes $z_r$. Assumption 2 is derived from Assumption 1 and an additional approximation that $e^x \approx 1 + x$. We make this approximation error concrete by introducing $\eta$, which we assume to be small. We show this derivation in the proof of Lemma 2.

**Definition 7** ($c_1$-high entropy). Let $T \subseteq [n]$ be the subset of tokens over which $p^i$ has significant probability. We say that $p^i$ has $c_1$-high entropy for $c_1 > 0$ if for all $t \in T$, $p_t^i \geq \frac{1}{c_1 |T|}$.

In other words, Definition 7 states that for tokens with significant probability under $p^i$, their probabilities are within a constant factor of $1/|T|$. That is, $p^i$ is somewhat close to uniform over $T$.

**Definition 8** ($c_2$-high quality). Let $T \subseteq [n]$ be the subset of tokens over which $p^i$ has significant probability. We say that $p^i$ has $c_2$-high quality for $c_2 > 0$ if for all $t \in T$, $q_t^i \geq p_t^i c_2$.

$c_2$-high quality states that the probabilities under the original model are within a constant factor of the probabilities under the adversary's model. This is true if the adversary makes bounded changes—recall that $p_t^i \approx q_t^i \cdot e^{(z_t - \vec{w}_t^*)}$. Therefore, it is true if $(z_t - \vec{w}_t^*) \leq -\log c_2$.

Observe that $c_1$-high entropy and $c_2$-high quality together imply the following fact:

**Fact 1.** *Let $p^i$ be $c_1$-high entropy and $c_2$-high quality, and let $T$ be the set of tokens with non-negligible probability under $p^i$. Then for all $t \in T$, $1/q_t^i \leq \frac{c_1 |T|}{c_2}$.*

**Lemma 2.** *Let $x_1, \ldots, x_k$ be any sequence of tokens generated using a model produced by the adversary. For all $i \in [k]$ with $c_1$-high entropy (Definition 7) and $c_2$-high quality (Definition 8), under reasonable assumptions (Assumptions 1 and 2), we have:*

$$\mathbb{E}[(\vec{w}_{\mathsf{wat}} - \vec{w}^*)_{x_i}] \geq c_1 \varepsilon^2 / c_2 - \alpha,$$

*where $\alpha = \frac{c_2 \eta \varepsilon \sqrt{2/\pi}}{c_1}$ is a very small approximation error term.*

We defer the proof of Lemma 2 to Appendix A.3.3.

**Assumption 3** (Independence). Consider a response output by the adversary's model, and let $x_1, \ldots, x_k$ be a subsequence of distinct tokens in this response. The random variables $(\vec{w}_{\mathsf{wat}} - \vec{w}^*)_{x_i}$ are independent.

This assumption is concerned with a degenerate case where the fact that a model output tokens with positive perturbation (e.g., $(\vec{w}_{\mathsf{wat}} - \vec{w}^*)_{x_i} > 0$) makes it less likely to output positive-perturbation tokens in the future. This is similar to an assumption used in Kirchenbauer et al. (2023) that the green list is freshly drawn for each token in the response; and an assumption used in Zhao et al. (2024) called "homophily".

Our main theorem states that if an adversary is given a watermarked model and it arbitrarily modifies its biases, the resulting model still produces watermarked text with overwhelming probability. This probability increases as $\varepsilon$ and the number of distinct tokens increase.

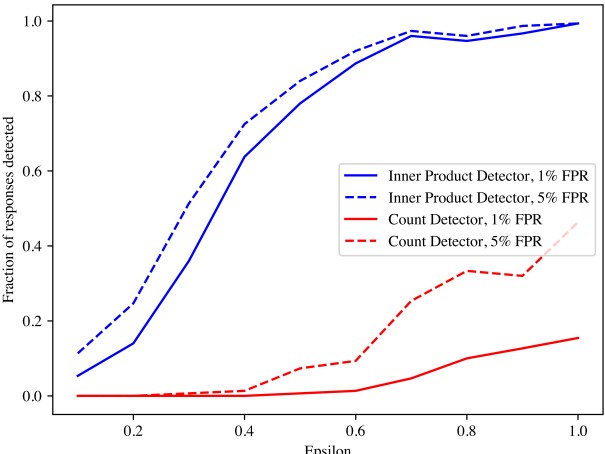

Figure 2: True positive detection rates for responses generated by OPT-1.3B, with our watermark applied under varying perturbation magnitudes (epsilon). The "Inner Product Detector" is our TextDetect detector (Algorithm 4), and the "Count Detector" is a baseline detector that we compare to.

**Theorem 3.** TextDetect *detects the watermark given an adversarially produced text with enough distinct tokens, enough entropy, and high enough quality.*

*More precisely, let $I$ be the $n \times n$ identity matrix, and let $\mathcal{C}$ be such that the adversary's posterior distribution over the original content $x$ after seeing $x_{\mathsf{wat}}$ is $\mathcal{N}(0, \varepsilon^2 I)$.*

*Let the adversary produce a model and a response generated by that model such that:*

- *The response contains at least $\lambda$ distinct tokens.*

- *At least a constant $\delta > 0$ fraction of the distributions $p^i$ of these distinct tokens are $c_1$-high entropy and $c_2$-high quality, for constants $c_1, c_2 > 0$. We let $\alpha = \frac{c_2 \eta \varepsilon \sqrt{2/\pi}}{c_1}$ be the corresponding approximation error term.*

*Under reasonable assumptions (Assumptions 1 to 3), TextDetect with threshold $\tau_{text} = \delta c_2 / 2 c_1 - \alpha / 2$ outputs* true *with overwhelming probability.*

**Theorem 4.** *For any constant threshold $\tau_{text} > 0$,* TextDetect *is sound.*

We defer the proofs of Theorems 3 and 4 to Appendices A.3.4 and A.3.5 respectively.

## 6 EXPERIMENTAL EVALUATION

In this section we present the results of an experimental evaluation of our watermarking construction which demonstrate its behavior under concrete parameter settings. Our experiments aim to show *feasibility* rather than optimality, as we open the door to further open-source watermarking work with experimental findings consistent with our theoretical results.

**Experimental Setup.** We watermark two LLMs: the OPT model with 1.3B and 6.7B parameters. We generate responses of three types: story, essay, and code; we include these prompts and additional prompt parameters in Appendix A.4. In all of our plots, we compute the fraction of responses detected among those with at least 20 distinct tokens, for a range of $\varepsilon$ parameter values. Recall that $\varepsilon$ is the standard deviation of the Gaussian perturbations added to the biases. Therefore, the detectability of the watermark increases with epsilon.

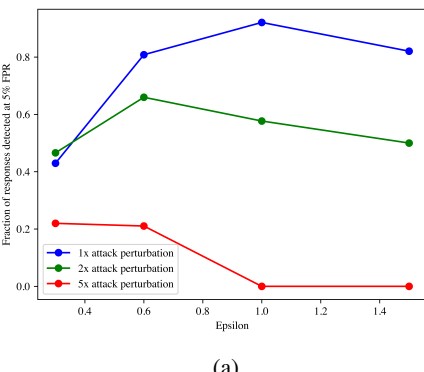 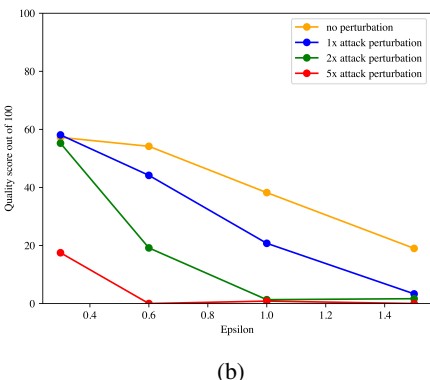

(a)                                                                              (b)

Figure 3: (a): Detection rates for adversarially perturbed watermarked OPT-6.7B models, to simulate a removal attack. The x-axis plots the epsilon parameter used in the watermark. The three curves show detection rates for models obtained by adding additional Gaussian noise to the biases, of standard deviaton 1, 2, and 5 times epsilon.
(b): Quality scores of watermarked texts generated with various parameters of epsilon, under the same perturbation attacks. The quality score of unwatermarked text was 59.933.

**Detectability.** In Figure 2, we plot the fraction of watermarked responses that are detected at 1% and 5% false positive rates. These responses are output by a model watermarked using our scheme, without further modification. The curves for "Inner Product Detector" show detectability under our detector TextDetect from Algorithm 4. Observe that for $\varepsilon \geq 0.5$, we have a true positive rate of at least 80%. Furthermore, this plot supports our result (Theorem 3) that the true positive rate increases with $\varepsilon$. We emphasize that although our detectability is not as strong as some inference-time watermarks, ours is the only *open-source* watermark with a provable unremovability guarantee.

We also plot detectability for an alternate detector ("Count Detector"). This alternate detector is analogous to that of Kirchenbauer et al. (2023); Zhao et al. (2024) in that it computes the fraction of tokens whose perturbations were positive (e.g., green list tokens). Observe that this detector has a significantly lower true positive rate than of TextDetect for our scheme. This shows that the open-source setting requires new techniques for reliable detection, and that existing inference-time detectors do not carry over even when implementable in the weights of the model. This supports our strategy of considering the magnitudes of the perturbations during detection, in our Inner Product Detector.

**Unremovability.** While we prove the unremovability properties of our watermark in Appendix A.3.4, in this section we consider a concrete attack adversary and show the unremovability-quality trade-off for it. The adversary produces a new model by adding Gaussian perturbations of mean zero and standard deviation 1x epsilon, 2x epsilon, or 5x epsilon. In Figure 3, we plot detection rates and quality scores for responses generated by adversarially modified watermarked models. That is, a watermarked model is produced at the given parameter epsilon shown on the x-axis.

In Figure 3a, we observe that in the 1x attack, for $\varepsilon \geq 0.6$ the detection rate remains above 80%. The watermark fails to persist in the larger-perturbation attacks in part because for large perturbations, the model produces highly repetitive responses, yielding few distinct tokens and a weak watermark signal. One can see this reflected in the quality scores shown in Figure 3b. The 5x attack, shown in red, has nearly zero quality for values of epsilon at least 0.6. The 2x attack, reaches zero quality at epsilon=1, at which point the watermark is still detectable with at least 50% probability.

In Figure 4, we show detection rates for watermarked responses subjected to a token substitution attack, as considered in Kuditipudi et al. (2024). We reproduce their attack, choosing a random subset of tokens in the response to substitute with uniform tokens from the token alphabet. We observe that our watermark tolerates moderate substitution attacks.

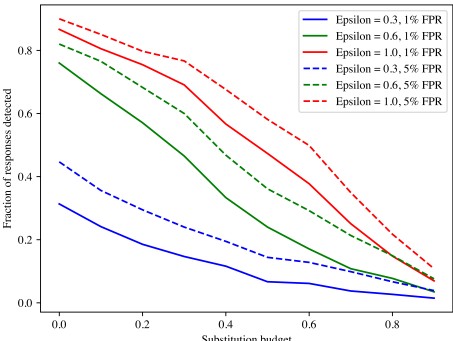

Figure 4: Detection rates for texts produced by a watermarked OPT-6.7B model and subjected to a substitution attack. The x-axis plots the fraction of tokens that are substituted. The curves show detection rates for varying epsilon parameters of the watermark.

**Additional discussion.** Our observed tradeoffs were affected by the phenomenon that large perturbations caused the model's responses to be very repetitive, therefore containing few distinct tokens. While higher values of epsilon (e.g., larger perturbations) increase the strength of our watermark, fewer distinct tokens *decrease* the strength of our watermark. This can be seen especially under the unremovability experiment where the adversary perturbs the model at 5x epsilon Figure 3. Here, the adversary's large perturbations caused the model to deteriorate so profoundly that most responses contained fewer than 20 distinct tokens, resulting in detection failure.

While our watermark is detectable in practice, and it exhibits an unremovability-quality tradeoff, the strong asymptotic tradeoff in Section 5 is somewhat dampened by the concrete parameters we consider. For example, the false negative rate is exponential in $-\varepsilon^4 T$, where $T$ is the number of distinct tokens. Although asymptotically this expression is negligible in the number of distinct tokens, for concrete values of $\varepsilon$ this term dominates. For example, suppose that the watermark was applied with $\varepsilon = 0.3$. We show that the false negative rate is at most $\exp(-(0.3)^4 T)$, where $T$ is the number of distinct tokens; to get any meaningful false positive guarantee we must have $T \geq (0.3)^{-4} \approx 125$. The vast majority of responses generated in our experiments had fewer than 100 distinct tokens, because we aimed to measure the detectability of paragraph-length texts of $\sim 300$ tokens.

**Quality.** In Figure 5, we plot quality scores for text generated by OPT-1.3B, using our watermark with various parameters of epsilon. We compute these quality scores by asking Mistral-Instruct-7B to provide a score out of 100. We report the average scores across 60 responses per value of epsilon, including responses of the three types we consider (essay, story, and code). *Unwatermarked* responses had an average quality score of 59.933.

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

# A APPENDIX

## A.1 CONCENTRATION BOUNDS

Here, we use a standard Gaussian vector of dimension $n$ to mean a vector that whose components are independent Gaussians with unit variance.

**Fact 3** (Theorem 2.7). *Let $\vec{x}$ be an iid standard Gaussian random vector of dimension $n$. For any $c \in (0, 1)$, we have*

$$\Pr\left[n(1 - c) < \|\vec{x}\|_2^2 < n(1 + c)\right] \geq 1 - 2\exp\left(-\frac{nc^2}{8}\right).$$

**Fact 4** (Gaussian tail bound). *Let $X \sim \mathcal{N}(0, \sigma^2)$. For any $t \geq 0$,*

$$\Pr[|X| \geq t] \leq 2\exp\left(\frac{-t^2}{2\sigma^2}\right).$$

**Fact 5** (Hoeffding's Inequality). *Let $X_1, \ldots, X_k$ be independent random variables in $[a, b]$, and let $X = \sum_{i=1}^{k} X_i$. Then for any constant $\delta \geq 0$,*

$$\Pr\left[|X - \mathbb{E}[X]| \geq \delta\right] \leq 2\exp\left(\frac{-\delta^2}{\sum_{i=1}^{k}(b-a)^2}\right).$$

## A.2 LANGUAGE MODELS

A *language model* operates over a token alphabet $\mathcal{T} = \{t_1, \ldots, t_n\}$. We sometimes identify tokens $t_i$ with their indices $i$. We consider autoregressive language models that apply the softmax function to obtain a probability distribution over the next token. That is, given a prompt and tokens output so far, the model computes a probability distribution $p = [p_1, \ldots, p_n]$ over the next token as follows. The last layer of the model computes logits $\ell_i, \ldots, \ell_n$ for each token. Each logit is computed as

$$\ell_i = w_{i,0} + \sum_{j=1}^{m} w_{i,j} v_{i,j}$$

where $w_{i,0}$ is the *bias* and $\sum_{j=1}^{m} w_{i,j} v_{i,j}$ is a weighted average of that node's inputs. The probability $p_i$ is computed by applying the softmax function to the logits; that is,

$$p_i = \frac{e^{\ell_i}}{\sum_{j=1}^{n} e^{\ell_j}}.$$

Because the probabilities are computed using the softmax function, no probabilities are exactly equal to zero; rather, they are extremely small. However, in practice these extremely small probabilities are functionally zero. We formally treat these small probabilities as cryptographically negligible.

## A.3 DEFERRED PROOFS

### A.3.1 PROOF OF THEOREM 1

*Proof.* In the removability game, let $\vec{w}^*$ denote the original content and $\vec{w}_{\mathsf{wat}}$ denote the watermarked content. By our assumption about the adversary's posterior distribution over $\vec{w}^*$, we have that from the adversary's perspective, $\vec{w}^* \sim \mathcal{N}(0, \varepsilon^2 I)$. We will show that for any $z$ produced by the adversary, either $z$ is far from $\vec{w}^*$, or $z$ is watermarked with high probability.

For any fixed $z \in \mathbb{R}^n$, letting $v = \vec{w}^* - \vec{w}_{\mathsf{wat}} \sim \mathcal{N}(0, \varepsilon^2 I)$ and $u = z - \vec{w}_{\mathsf{wat}}$,

$$(z - \vec{w}^*) \cdot (\vec{w}_{\mathsf{wat}} - \vec{w}^*) = (v - u) \cdot v.$$

We now argue that the quantity $(v - u) \cdot v$ is likely to be large. Let $v_1$ denote the first component of $v$, let $\delta = \frac{1}{\sqrt{n}}\|u\|_2$, and let $f > 1$ be a constant.

$$
\begin{aligned}
\Pr_{v \leftarrow \mathcal{N}(0,\varepsilon^2 I)}[(v - u) \cdot v < \tau\varepsilon^2 n] &= \Pr_{v \leftarrow \mathcal{N}(0,\varepsilon^2 I)}\left[\|v\|_2^2 - \delta v_1 \sqrt{n} < \tau\varepsilon^2 n\right] \\
&\leq \Pr_{v \leftarrow \mathcal{N}(0,\varepsilon^2 I)}\left[\|v\|_2^2 - \delta\varepsilon f\sqrt{n} < \tau\varepsilon^2 n \text{ and } v_1 \leq \varepsilon f\right] + \Pr_{v \leftarrow \mathcal{N}(0,\varepsilon^2 I)}[v_1 > \varepsilon f] \\
&\leq \Pr_{v \leftarrow \mathcal{N}(0,\varepsilon^2 I)}\left[\|v\|_2^2 - \delta\varepsilon f\sqrt{n} < \tau\varepsilon^2 n\right] + \frac{1}{2\pi}\exp\left(-f^2/2\right) \\
&= \Pr\left[\varepsilon^2\|\mathcal{N}(0,I)\|_2^2 < \tau\varepsilon^2 n + \delta\varepsilon f\sqrt{n}\right] + \frac{1}{2\pi}\exp\left(-f^2/2\right) \\
&= \Pr\left[\|\mathcal{N}(0,I)\|_2^2 < \left(\tau + \frac{\delta f}{\varepsilon\sqrt{n}}\right)n\right] + \frac{1}{2\pi}\exp\left(-f^2/2\right) \\
&\leq 2\exp\left(-n\left[1 - \tau - \frac{\delta f}{\varepsilon\sqrt{n}}\right]^2/8\right) + \frac{1}{2\pi}\exp\left(-f^2/2\right) \qquad (1)
\end{aligned}
$$

Above, we made use of the facts that $v \sim \mathcal{N}(0,\varepsilon^2 I) \sim \varepsilon\mathcal{N}(0,I)$, and $v_1 \sim \mathcal{N}(0,\varepsilon^2)$. We also made use of the fact that for all $\ell \in \mathbb{R}$ and $u \in \mathbb{R}^n$,

$$
\Pr_{v \leftarrow \mathcal{N}(0,\varepsilon^2 I)}[u \cdot v = \ell] = \Pr_{v_1 \leftarrow \mathcal{N}(0,\varepsilon^2)}[v_1\|u\|_2 = \ell],
$$

which follows from spherical symmetry of a Gaussian.

We've now upper bounded the probability that $(v - u) \cdot v$ is large in Equation (1); it remains to analyze this probability bound. Observe that for any $f = \omega(\sqrt{\log n})$, in Equation (1), $\frac{1}{2\pi}e^{-f^2/2}$ is negligible. For any $\delta = o(\varepsilon n/f) = o(\varepsilon n/\sqrt{\log n})$, in Equation (1) we have that $\tau n - \frac{\delta f\sqrt{n}}{\varepsilon}$ is at least $(1+c)n$ for some constant $c = \tau - 1 - o(1)$. By Fact 3, $\Pr\left[\|\mathcal{N}(0,I)\|_2^2 \leq \tau n - \frac{\delta f\sqrt{n}}{\varepsilon}\right] \leq \mathsf{negl}(n)$. Therefore, the expression in Equation (1) is negligible in $n$.

Applying the above, if $z$ is unwatermarked with non-negligible probability we must have $\|z - \vec{w}_{\mathsf{wat}}\|_2 \geq \omega\left(\frac{\varepsilon n}{\sqrt{\log n}}\right)$. By Fact 3, for any constant $\alpha$, $\Pr\left[\|\vec{w}_{\mathsf{wat}} - \vec{w}^*\|_2 \geq \frac{\alpha\varepsilon n}{\sqrt{\log n}}\right] \leq \mathsf{negl}(n)$. By a union bound, if $z$ is unwatermarked then with overwhelming probability we must have $\|z - \vec{w}_{\mathsf{wat}}\|_2 \geq \omega\left(\frac{\varepsilon n}{\sqrt{\log n}}\right)$ and $\|\vec{w}_{\mathsf{wat}} - \vec{w}^*\|_2 < \frac{\alpha\varepsilon n}{\sqrt{\log n}}$. By a triangle inequality,

$$
\begin{aligned}
\|z - \vec{w}^*\|_2 &\geq \|z - \vec{w}_{\mathsf{wat}}\|_2 - \|\vec{w}_{\mathsf{wat}} - \vec{w}^*\|_2 \\
&\geq \omega\left(\frac{\varepsilon n}{\sqrt{\log n}}\right) - \frac{\alpha\varepsilon n}{\sqrt{\log n}} \\
&\geq \omega\left(\frac{\varepsilon n}{\sqrt{\log n}}\right).
\end{aligned}
$$

Finally, again by Fact 3, with overwhelming probability $\|\vec{w}_{\mathsf{wat}} - \vec{w}^*\|_2 = \Theta(\varepsilon\sqrt{n})$. Therefore, $\|z - \vec{w}^*\|_2 \geq \omega\left(\frac{\sqrt{n}}{\sqrt{\log n}}\right)$.

$\square$

### A.3.2 PROOF OF THEOREM 2

*Proof.* Let $x' \in \mathbb{R}^n$. We will show that with overwhelming probability, $\|x' - x_{\mathsf{wat}}\| > \frac{1}{2}\varepsilon n$. Since the detector outputs true only if $\|x' - x_{\mathsf{wat}}\| > \frac{1}{2}\varepsilon n$, this is sufficient to show that any fixed $x'$ will not be falsely detected.

Let $\Delta = x_{\mathsf{wat}} - x$, and observe that

$$
\begin{aligned}
\|x' - x_{\mathsf{wat}}\|^2 &= \|x' - (x + \Delta)\|^2 \\
&= \|\Delta - (x - x')\|^2 \\
&= \|\Delta\|^2 - 2\Delta \cdot (x - x') + \|x - x'\|^2
\end{aligned}
$$

Recall that by spherical symmetry of a Gaussian, $\Delta \cdot (x - x')$ is distributed as $\Delta_1 \|x - x'\|$ where $\Delta_1 \sim \mathcal{N}(0, \varepsilon^2)$. Therefore, with overwhelming probability, $2\Delta(x - x') \leq 2\sqrt{n}\|x - x'\|$. By Fact 3 and a union bound, with overwhelming probability we also have $\|\Delta\|^2 \geq 0.9\varepsilon^2 n^2$. Therefore,

$$\|x' - x_{\mathsf{wat}}\|^2 \geq 0.9\varepsilon^2 n^2 + \|x - x'\|(\|x - x'\| - 2\sqrt{n})$$

We now consider two cases. If $\|x - x'\| \geq 2\sqrt{n}n$, we have $\|x' - x_{\mathsf{wat}}\|^2 \geq 0.9\varepsilon^2 n^2$; therefore, $\|x' - x_{\mathsf{wat}}\| \geq \frac{1}{2}\varepsilon n$ as desired. If $\|x - x'\| < 2\sqrt{n}$, we have $\|x' - x_{\mathsf{wat}}\|^2 \geq 0.9\varepsilon^2 n^2 + 4n \geq 0.8\varepsilon^2 n^2$ for sufficiently large $n$. $\qquad\square$

### A.3.3 PROOF OF LEMMA 2

*Proof.* We first show that Assumption 1 implies a useful relationship between $(z_t - \vec{w}_t^*)$ and the ratio between $p_t^i$ and $q_t^i$. Observe that

$$
\begin{aligned}
p_t^i &= \frac{e^{\ell_t^i + z_t}}{\sum_{r \in [n]} e^{\ell_r^i + z_r}} \\
&= \frac{e^{\ell_t^i + \vec{w}_t^* + (z_t - \vec{w}_t^*)}}{\sum_{r \in [n]} e^{\ell_r^i + z_r}} \\
&\approx \frac{e^{\ell_t^i + \vec{w}_t^* + (z_t - \vec{w}_t^*)}}{\sum_{r \in [n]} e^{\ell_r^i + \vec{w}_r^*}} \text{ by Assumption 1} \\
&= \frac{e^{\ell_t^i + \vec{w}_t^*}}{\sum_{r \in [n]} e^{\ell_r^i + \vec{w}_r^*}} \cdot e^{(z_t - \vec{w}_t^*)} \\
&= q_t^i \cdot e^{(z_t - \vec{w}_t^*)} \\
&\approx q_t^i + q_t^i \cdot (z_t - \vec{w}_t^*) \text{ using the approximation that } e^x \approx 1 + x.
\end{aligned}
$$

Rearranging, we have the useful fact:

$$(z_t - \vec{w}_t^*) \approx p_t^i / q_t^i - 1$$

where $\approx$ comes from Assumption 1 and the approximation $e^x \approx 1 + x$. Thus we have arrived at Assumption 2.

Armed with Assumptions 1 and 2, and Fact 1, we can prove the lemma. Our goal is to rewrite the inner product in terms of the expectation of $(\vec{w}_{\mathsf{wat}} - \vec{w}^*)_r$ for tokens $r$ appearing in the given text. Let $T$ denote the set of tokens over which $p^i$ has non-negligible probability. Let $p^i|_T$ denote the vector of probabilities when $p^i|_T$ is restricted only to tokens in $T$. Let $(z - \vec{w}^*)|_T$ and $(\vec{w}_{\mathsf{wat}} - \vec{w}^*)|_T$ denote the vectors $(z - \vec{w}^*)$ and $(\vec{w}_{\mathsf{wat}} - \vec{w}^*)$ restricted to the tokens in $T$.

We first manipulate the expression for the dot product and apply Assumption 2:

$$
\begin{aligned}
(z - \vec{w}^*)|_T \cdot (\vec{w}_{\mathsf{wat}} - \vec{w}^*)|_T &= \sum_{t \in T} (z_t - \vec{w}_t^*) \cdot (\vec{w}_{\mathsf{wat}} - \vec{w}^*)_t \\
&\leq \sum_{t \in T} (p_t^i / q_t^i - 1) \cdot (\vec{w}_{\mathsf{wat}} - \vec{w}^*)_t + \eta |(\vec{w}_{\mathsf{wat}} - \vec{w}^*)_t| \text{ by Assumption 2} \\
&= \mathbb{E}_{r \sim p^i|_T} \left[ \frac{1}{q_r^i} (\vec{w}_{\mathsf{wat}} - \vec{w}^*)_r \right] - \sum_{t \in T} (\vec{w}_{\mathsf{wat}} - \vec{w}^*)_t + \eta |(\vec{w}_{\mathsf{wat}} - \vec{w}^*)_t|
\end{aligned}
$$

(2)

We now analyze the expression $\mathbb{E}_{r \sim p^i|_T} \left[ \frac{1}{q_r^i} (\vec{w}_{\mathsf{wat}} - \vec{w}^*)_r \right]$ to remove the term $\frac{1}{q_r^i}$, which cannot be determined from just the text. The high entropy and quality of $p^i$ lets us do exactly this, invoking Fact 1.

Recall that Fact 1 says that for the set of tokens $T$ over which $p^i$ has non-negligible probability, for all $t \in p^i$ we have $1/q_t^i \leq c_1 |T|/c_2$. Let $p^i|_T$ denote $p^i$ restricted to tokens in $T$ and renormalized. Applying this inequality, we have

$$\mathop{\mathbb{E}}_{r\sim p^i|_T}\left[\frac{1}{q_r^i}(\vec{w}_{\mathsf{wat}}-\vec{w}^*)_r\right] = \sum_{t\in T}p_t^i(1/q_t^i)(\vec{w}_{\mathsf{wat}}-\vec{w}^*)_t$$

$$\leq \frac{c_1|T|}{c_2}\sum_{t\in T}p_t^i(\vec{w}_{\mathsf{wat}}-\vec{w}^*)_t \text{ by Fact 1}$$

$$= \frac{c_1|T|}{c_2}\mathop{\mathbb{E}}_{r\sim p^i|_T}\left[(\vec{w}_{\mathsf{wat}}-\vec{w}^*)_r\right]$$

Therefore,

$$\mathop{\mathbb{E}}_{r\sim p^i|_T}\left[(\vec{w}_{\mathsf{wat}}-\vec{w}^*)_r\right] \geq \frac{c_2}{c_1|T|}\mathop{\mathbb{E}}_{r\sim p^i|_T}\left[\frac{1}{q_r^i}(\vec{w}_{\mathsf{wat}}-\vec{w}^*)_r\right]$$

$$\geq \frac{c_2}{c_1|T|}\left((z-\vec{w}^*)|_T\cdot(\vec{w}_{\mathsf{wat}}-\vec{w}^*)|_T + \sum_{t\in T}(\vec{w}_{\mathsf{wat}}-\vec{w}^*)_t\right) - \eta|(\vec{w}_{\mathsf{wat}}-\vec{w}^*)_t| \text{ by Equation (2)}$$

which is a lower bound on this expected value with no dependence on $q$, as desired.

Finally, we analyze the expected value over the choice of $\vec{w}^*$. Recall that from the adversary's perspective, $\vec{w}^* \sim \mathcal{N}(0,\varepsilon^2 I)$. This immediately lets us simplify the term with $\eta$, using the mean of the folded normal distribution:

$$\mathop{\mathbb{E}}_{\vec{w}^*}\left[\eta|(\vec{w}_{\mathsf{wat}}-\vec{w}^*)_t|\right] = \eta\varepsilon\sqrt{2/\pi}. \tag{3}$$

As in the proof of Theorem 1, let $v = \vec{w}^* - \vec{w}_{\mathsf{wat}}$ and let $u = z - \vec{w}_{\mathsf{wat}}$. Then $(z-\vec{w}^*)\cdot(\vec{w}_{\mathsf{wat}}-\vec{w}^*) = (u-v)\cdot v$, where $v\sim\mathcal{N}(0,\varepsilon^2 I)$.

$$\mathop{\mathbb{E}}_{\vec{w}^*}\mathop{\mathbb{E}}_{r\sim p^i|_T}\left[(\vec{w}_{\mathsf{wat}}-\vec{w}^*)_r\right] \geq \mathop{\mathbb{E}}_{\vec{w}^*}\left[\frac{c_2}{c_1|T|}\left((z-\vec{w}^*)|_T\cdot(\vec{w}_{\mathsf{wat}}-\vec{w}^*)|_T + \sum_{t\in T}(\vec{w}_{\mathsf{wat}}-\vec{w}^*)_t - \eta|(\vec{w}_{\mathsf{wat}}-\vec{w}^*)_t|\right)\right]$$

$$= \mathop{\mathbb{E}}_{\vec{w}^*}\left[\frac{c_2}{c_1|T|}\left((z-\vec{w}^*)|_T\cdot(\vec{w}_{\mathsf{wat}}-\vec{w}^*)|_T + \sum_{t\in T}(\vec{w}_{\mathsf{wat}}-\vec{w}^*)_t\right)\right] - \frac{c_2\eta\varepsilon\sqrt{2/\pi}}{c_1} \text{ by Equation (3)}$$

$$= \mathop{\mathbb{E}}_{\vec{w}^*}\left[\frac{c_2}{c_1|T|}(z-\vec{w}^*)|_T\cdot(\vec{w}_{\mathsf{wat}}-\vec{w}^*)|_T\right] - \frac{c_2\eta\varepsilon\sqrt{2/\pi}}{c_1} \text{ since each } (\vec{w}_{\mathsf{wat}}-\vec{w}^*)_t \text{ has mean 0}$$

$$= \mathop{\mathbb{E}}_{v\sim\mathcal{N}(0,\varepsilon^2 I)}\left[\frac{c_2}{c_1|T|}(u-v)|_T\cdot v|_T\right] - \frac{c_2\eta\varepsilon\sqrt{2/\pi}}{c_1}$$

$$= \mathop{\mathbb{E}}_{v\sim\mathcal{N}(0,\varepsilon^2 I)}\left[\frac{c_2}{c_1|T|}\left(\|v|_T\|_2^2 - u|_T\cdot v|_T\right)\right] - \frac{c_2\eta\varepsilon\sqrt{2/\pi}}{c_1}$$

$$= \mathop{\mathbb{E}}_{v\sim\mathcal{N}(0,\varepsilon^2 I)}\left[\frac{c_2}{c_1|T|}\|v|_T\|_2^2\right] - \frac{c_2\eta\varepsilon\sqrt{2/\pi}}{c_1} \text{ since each } u_i\cdot v_i \text{ has mean 0}$$

$$= c_2\varepsilon^2/c_1 - \frac{c_2\eta\varepsilon\sqrt{2/\pi}}{c_1}$$

Therefore,

$$\mathop{\mathbb{E}}_{\substack{\vec{w}^* \\ r\sim p^i|_T}}\left[\vec{w}_{\mathsf{wat}}-\vec{w}^*\right] \geq c_2\varepsilon^2/c_1 - \frac{c_2\eta\varepsilon\sqrt{2/\pi}}{c_1}.$$

$\square$

### A.3.4 PROOF OF THEOREM 3

.

*Proof.* First, observe that $\mathbb{E}_{x_i \sim p_i}[(\vec{w}_{\mathsf{wat}} - \vec{w}^*)_{x_i}] \geq 0$ for any distribution $p_i$ produced by the adversary. We've shown in Lemma 2 that for $p_i$'s that are $c_1$-high entropy and $c_2$-high quality, $\mathbb{E}_{x_i \sim p_i|_T}[(\vec{w}_{\mathsf{wat}} - \vec{w}^*)_{x_i}] \geq c_2 \varepsilon^2 / c_1$, where $T$ is the set of tokens with non-negligible probability. Formally, we model the tokens appearing in the text as sampled from $p^i|_T$ rather than $p^i$, since the negligible probability tokens will never be sampled.

By assumption, at least a $\delta$ fraction of $p_i$'s have are $c_1$-high entropy and $c_2$-high quality. Therefore, letting $U$ denote the set of indices of distinct tokens in the response,

$$\frac{1}{|U|} \mathbb{E} \left[ \sum_{i \in U} (\vec{w}_{\mathsf{wat}} - \vec{w}^*)_{x_i} \right] \geq \frac{\delta c_2 \varepsilon^2}{c_1} - \alpha = 2\varepsilon^2 \tau_{\mathsf{text}}$$

By Assumption 3, the $(\vec{w}_{\mathsf{wat}} - \vec{w}^*)_{x_i}$'s are independent. Furthermore, all $(\vec{w}_{\mathsf{wat}} - \vec{w}^*)_{x_i}$'s lie in $[-\lambda^{1/4}, \lambda^{1/4}]$ with overwhelming probability by a standard Gaussian tail bound (Fact 4):

$$\Pr \left[ \exists x_j \in U \text{ s.t. } \Delta(x_j) \notin [-\lambda^{1/4}, \lambda^{1/4}] \right] \leq 2|U| \exp \left( \frac{-\lambda^{1/2}}{2\varepsilon^2} \right).$$

Let $X = \sum_{i \in U} (\vec{w}_{\mathsf{wat}} - \vec{w}^*)_{x_i}$. By a Hoeffding bound (Fact 5),

$$\Pr \left[ |X - \mathbb{E}[X]| \geq \mathbb{E}[X]/2 \right] \leq 2 \exp \left( \frac{-2(\mathbb{E}[X]/2)^2}{|U|(2\lambda^{1/4})^2} \right)$$

$$\leq 2 \exp \left( \frac{-2|U|^2 \varepsilon^4 \tau_{\mathsf{text}}^2}{4|U|\sqrt{\lambda}} \right) \text{ since } \mathbb{E}[X] \geq |U|2\varepsilon^2 \tau_{\mathsf{text}}$$

$$= 2 \exp \left( \frac{-|U|\varepsilon^4 \tau_{\mathsf{text}}^2}{2\sqrt{\lambda}} \right)$$

which is negligible in $\lambda$, as $|U| \geq \lambda$. $\square$

### A.3.5 PROOF OF THEOREM 4

*Proof.* Consider a given text, and recall that our detector only looks at distinct tokens. Therefore, let $x_1, \ldots, x_k$ denote the given text with the duplicate tokens removed. Observe that for each $i \geq \lambda$, the count during that iteration is $\sum_{j=1}^{i} \Delta(x_j)$, where each $\Delta(x_j)$ is an i.i.d. Gaussian $\mathcal{N}(0, \varepsilon^2)$. Therefore, their sum is also a Gaussian random variable $X \sim \mathcal{N}(0, i\varepsilon)$.

By Fact 4,

$$\Pr \left[ X \geq i\tau_{\mathsf{text}}\varepsilon^2 \right] \leq 2 \exp \left( \frac{-2i^2 \tau_{\mathsf{text}}^2 \varepsilon^4}{2i\varepsilon^2} \right)$$

$$= 2 \exp \left( -i\tau_{\mathsf{text}}^2 \varepsilon^2 \right)$$

Since $i \geq \lambda$, this is negligible in $\lambda$. By a union bound, the probability that this sum exceeds the threshold for any $i$ is at most $k$ times this expression, which is also negligible. $\square$

### A.4 ADDITIONAL EXPERIMENT SPECIFICATIONS AND FIGURES

**Experiment specifications.** We generate responses of up to 300 tokens using random sampling, at a temperature of 0.9. We set the maximum ngram repeat length to 5, to avoid overly repetitive responses. We generate three types of responses: story responses, essay responses, and code responses. We use the following prompts:

**Story prompt:** "Here is one of my favorite stories: It was a "

**Essay prompt:** "Here is one of my favorite essays: It is often thought that "

**Code prompt:** "Here is a python script for your desired functionality: import "

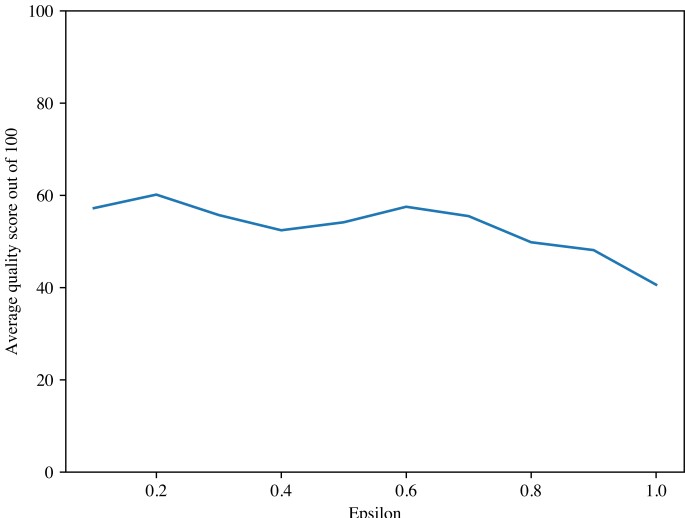

Figure 5: Quality scores of watermarked texts generated with various parameters of epsilon. The quality score of unwatermarked text was 59.933.

To evaluate quality, we ask Mistral-7B-Instruct to evaluate each response based on its type, and to provide a score out of 100.

**Additional figures.** We include an additional figure showing the quality of watermarked texts. We also include larger versions of Figure 3.

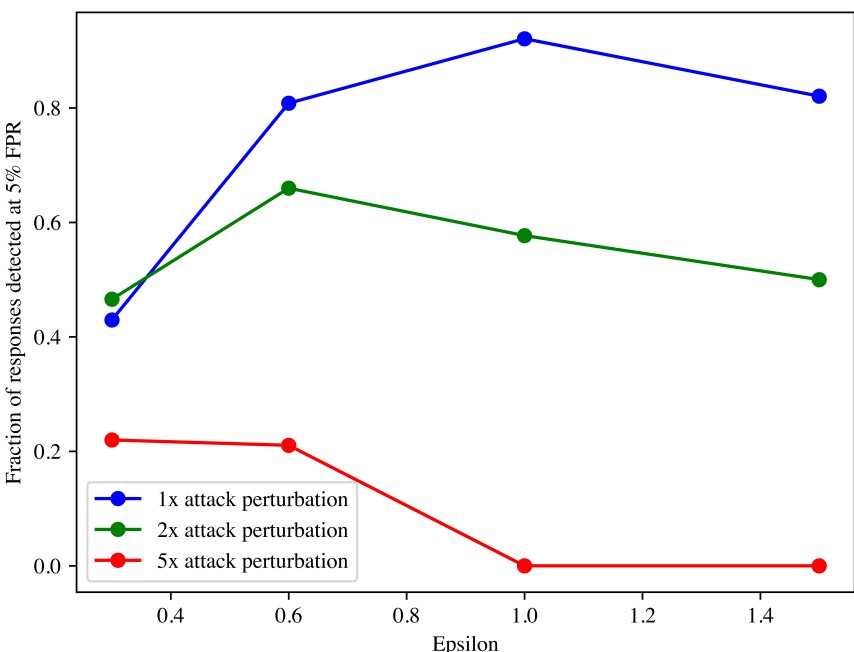

Figure 6: Detection rates for adversarially perturbed watermarked OPT-6.7B models, to simulate a removal attack. The x-axis plots the epsilon parameter used in the watermark. The three curves show detection rates for models obtained by adding additional Gaussian noise to the biases, of standard deviaton 1, 2, and 5 times epsilon. (Figure 3a)

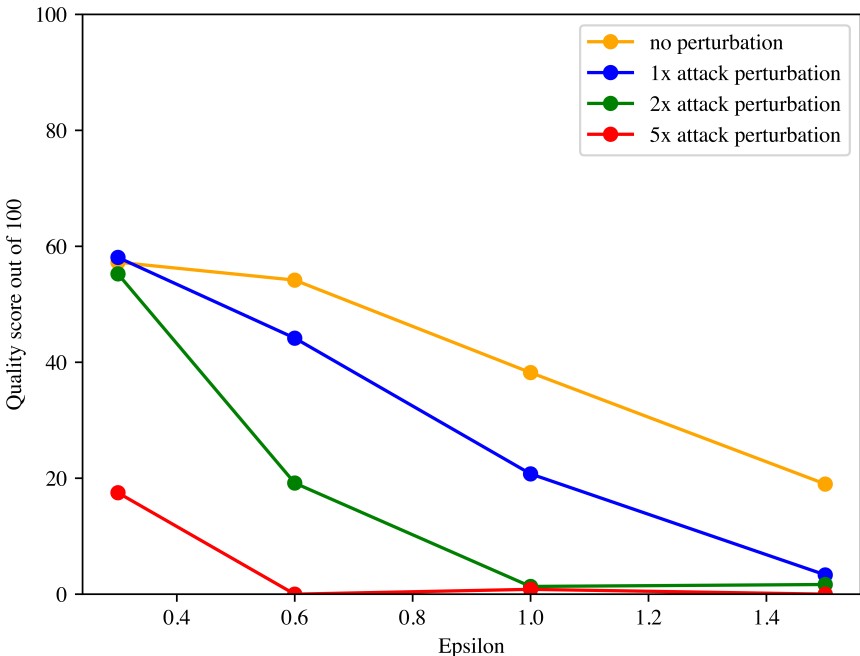

Figure 7: Quality scores of watermarked texts generated with various parameters of epsilon, under the same perturbation attacks. The quality score of unwatermarked text was 59.933. (Figure 3b)

