# OpenReview forum: "Unremovable Watermarks for Open-Source Language Models"
_ICLR.cc/2025/Conference — Submitted to ICLR 2025_

### Official Review · Reviewer_p4oF · 2024-10-20

**Soundness:** 2
**Presentation:** 2
**Contribution:** 2
**Rating:** 6
**Confidence:** 3

**Summary:**

The problem of watermarking the output of LLMs has been around for some time. Previous work has focused on changing the output distribution of the LLMs, sometimes in an “undetectable/distortion-free” way. But this work starts by making the following point:
If one has access to the parameters of an LLM, they can run it to generate output that is not watermarked.

The main problem of this paper is to watermark the parameters of the model itself, in a way that it both gets reflected in the output + even if one gets their hand on the model parameters, they cannot modify it in a way that the watermark is removed from subsequent outputs.

Of course one shall consider attackers who can train the model from scratch. So the paper assumes that doing so is hard (e.g., by making the access to the data generation process costly).

The contribution of the paper is the following. They propose a scheme that works by adding Gaussian noise to the last layer of the model before publishing it. Also knowing the original model and the noise, they show how to verify whether the generated output comes from their model or not.

The paper then makes multiple assumptions to prove that their scheme is secure. The final watermark suffers from not having robustness or undetectability. It is not clear if such weaknesses are inherent or not.

**Strengths:**

As far as I know, this is the first work that aims to formally define and address “unremovable watermarks” that are planted in open source models.

**Weaknesses:**

The paper does not fully address several, by now well-recognized aspects, of the watermark:
1. Robustness of the watermarks. E.g., what if one changes even two characters of the produced output? Or that it deletes parts of the output. Here the paper claims it has done experiments but i could not figure out what exact perturbation channels are studied.
2. It seems that the output of the watermarked model here is *not* indistinguishable -- sometimes called undetectable or distortion free -- (in comparison with non-watermarked model's output). This is the ultimate way of arguing that the model’s utility does not degrade after adding the watermark and the paper does not discuss it clearly. Note that here, I am not talking about "removability". This is about the item above (robustness) but rather if the output of the watermarked model differs (in a computationally noticeable way) from the output of non-watermarked model.

To partially address the above issues, the paper should first define clearly what class of perturbation channels they study (and why they are interesting) for the robustness property evolutions (which are seemingly done under the name of Detectability) and for the item 2 above (undetectability of the output -- which is different from the Detectability study) they should design experiments specifically for this goal or make a theoretical assertion.

Also, the proofs are based on multiple assumptions, which make the final conclusion far from ideal. (See my question below)

Also, what happens to the watermarks if the model is fine tuned? (note that black-box methods still work, if the model is fine tuned). This issue should be addressed using experiments. they could be simple experiments that simply test the detectability and utility of the outputs after a fine tuning for specific goals (also see my question below).

The writing also is not great and lacks discussions and justifications with regard to the issue mentioned above (e.g., of the assumptions).
Other than the issues above, the intuition behind why this non-black-box approach is working could be much better.

Other minor comments on writing:

Def 2 seems to be more like a “similarity” measure, because the loss in quality seems to be different. For example, two models could look very different but have the same quality of responses.

Def 4: seems to mix the input space of Q and \ell, right?

**Questions:**

Can you address the two issues of robustness (to small change in the output) and the undetectability of the output (compared to the non-watermarked model) ?

 In your experiments, how do you measure that the utility of the model has not degraded after adding the watermark. I know that you have an oracle that measures the degrading, but then you instantiate the oracle using mathematical formulas regarding the model. But how do you make sure that this reflects the actual quality of the model’s output? For example, you could use specific metrics or human evaluation methods to assess output quality more directly.

Can you discuss why the assumptions are fine? There are 3 explicit assumptions and (multiple) implicit assumptions in the statement of Theorem 1 (eg., “let C be such that…” or “c_2-high quality…”) I think that discussion is needed before calling assumptions reasonable (instead of putting the word reasonable in the theorem statement).

Can you argue either way about the effect of fine tuning in your watermarked model?

In your experiments: can you be more explicit about what your attacker is? e.g., using a pseudocode.

---

> ### Author Response · Authors · 2024-11-25
>
> > Can you address the two issues of robustness (to small change in the output) and the undetectability of the output (compared to the non-watermarked model) ?
>
> We prove that the watermark is robust to an adversary that changes the weights of the model; this is why we use the term “unremovable” rather than “robust,” which typically refers to an adversary that changes the response but not the model. We show empirically but do not prove that our watermark is robust to substitution attacks (Figure 4). We agree that showing how unremovability translates to robustness would strengthen the paper. Our scheme is not distortion-free or undetectable. Like Zhao et al. and Kirchenbauer et al., we instead prove a bound on the amount that our watermark changes the output distribution; our proof is via our quality notion. This fact is buried in the proof of Theorem 1 in the appendix; we will emphasize it in the main body. We also show empirically how quality is affected by the watermark, in Figure 3b.
>
> > In your experiments, how do you measure that the utility of the model has not degraded after adding the watermark. I know that you have an oracle that measures the degrading, but then you instantiate the oracle using mathematical formulas regarding the model. But how do you make sure that this reflects the actual quality of the model’s output? For example, you could use specific metrics or human evaluation methods to assess output quality more directly.
>
> In our experiments, we use Mistral-7B-Instruct as a quality oracle, following similar work (Piet et al.).
>
> > Can you discuss why the assumptions are fine? There are 3 explicit assumptions and (multiple) implicit assumptions in the statement of Theorem 1 (eg., “let C be such that…” or “c_2-high quality…”) I think that discussion is needed before calling assumptions reasonable (instead of putting the word reasonable in the theorem statement).
>
> We discuss the assumptions when they are introduced, in Section 5.2 before Theorem 1. If you have already seen this, can you please be more specific about your concerns?
>
> > Can you argue either way about the effect of fine tuning in your watermarked model?
>
> We will add discussion about this. We show that removing the watermark either requires knowledge about the distribution of high-quality text, or results in a model with degraded quality. Formally, our result implies that if fine tuning alters only the last layer of the model, it either fails to remove the watermark, or it uses knowledge of the distribution of high-quality text (e.g., via high-quality training data).
>
> > In your experiments: can you be more explicit about what your attacker is? e.g., using a pseudocode.
>
> Yes, we will add pseudocode. We also describe the attacks simply here: In Figures 3a and b, the attacker adds noise to the last layer biases where the noise added to each bias is drawn independently from distributions $N(0, \epsilon^2)$ (1x attack perturbation), from $N(0, (2\epsilon)^2)$ (2x attack perturbation), and $N(0, (5\epsilon)^2)$ (5x attack perturbation) respectively. In Figure 4, the attacker randomly selects a subset of tokens in the response, of the size given on the x axis (the substitution budget). It then substitutes them with random tokens.

---

> ### Comment · Reviewer_p4oF · 2024-11-26
> **Re:**
>
> Thanks for the clarifications.
>
> Re assumptions: I am not claiming them to be unreasonable; I just cannot get a good sense of them by reading the (sometimes minimal) discussions. E.g., I still don't understand Assumption 1's discussions, and I don't understand how the restrictions on C in Theorem 1 should be interpreted. you mentioned that this is discussed in Section 5.2 before Theorem 1, but Theorem 1 is before Section 5.2. If you give a more accurate pointer, I will double check.
>
> Also, I wanted to comment on something that you seem to be commenting on it yourself first. As I understand, un-removability is *not* a strictly stronger property than black-box robustness, right? Namely, if we prove a model to be non-black-box robust, it is not clear what kind of black-box robustness properties we can infer? That makes the formulation of un-removability a bit less appealing, but I have no suggestions on how to fix it, so I do not take it against the paper.
>
> All in all, I find the problem if this paper very interesting, but the presentation makes it a bit hard to judge how much actual of a jump is made towards this new (interesting) direction.
>
> In summary, I am a bit more positive about the paper now and increased my score.

---

### Official Review · Reviewer_p6Eu · 2024-10-28

**Soundness:** 2
**Presentation:** 2
**Contribution:** 1
**Rating:** 3
**Confidence:** 3

**Summary:**

This paper introduces the first watermarking scheme for open-source LLMs. The idea is to embed a watermark directly in the model's parameters rather than through the sampling algorithm, making it resilient to tampering in open-source environments. The authors define "unremovable watermarks" for neural networks and implement a scheme that perturbs neuron biases in the model's final layer with Gaussian noise. Detection of the watermark is achieved either by examining the weights for specific bias patterns or by analyzing output text for token frequency markers. The watermark is shown to be unremovable, as attempts to erase it degrade model quality, with experimental results on OPT-6.7B and OPT-1.3B supporting this claim.

**Strengths:**

- The paper introduces a new watermarking scheme to embed unremovable watermarks directly in model weights and resist tampering in open-source environments.
- The paper defines "unremovable watermarks," providing proofs and an analysis of the watermark’s robustness against attacks and conducting experiments with OPT-6.7B and OPT-1.3B models to demonstrate the approach's effectiveness.
- The paper is well-structured, logically presenting its motivation, methodology, and findings, with clear definitions and algorithms. I highly commend the authors for the nice presentation.

**Weaknesses:**

- Now, the authors claim to have introduced the first watermarking scheme for open-source LLMs. What do they mean by this? There are many watermarking schemes which could be deployed in open source LLMs, so this claim might not be right, as the proposed scheme can also be deployed in closed source LLMs by the model owners. Which leads to the next question. If the LLM is open source, what exactly is the benefit of watermarking when the attacker has direct access to model's weights. Can the authors expand more on their motivation?
- The proposed approach embeds watermark signals to the bias of the last layer's neurons. There is another approach by ByteDance that injects watermark into the LLM weights by finetuning (https://arxiv.org/pdf/2403.10553). Why is there no comparison with this approach? Infact, why is there no comparison with other watermarking schemes at all?
- There are adaptive ways to bypass watermarks. One is by using adaptive paraphrasers. If the proposed watermark scheme is unremovable, yet detectable, why are there no empirical results proving the 'unremovability' claim using adaptive paraphrasers, or even normal paraphrasers like Dipper, or even using open source LLMs for paraphrasing.
- How efficient is the detection process? How many tokens does it require to detect the proposed scheme, especially using its optimal hyperparameters? I feel the experiments the authors provided to prove the efficiency and strength of this approach are not enough.

**Questions:**

Please answer those in weaknesses above.

---

> ### Author Response · Authors · 2024-11-25
>
> > Now, the authors claim to have introduced the first watermarking scheme for open-source LLMs. What do they mean by this? There are many watermarking schemes which could be deployed in open source LLMs, so this claim might not be right, as the proposed scheme can also be deployed in closed source LLMs by the model owners. Which leads to the next question. If the LLM is open source, what exactly is the benefit of watermarking when the attacker has direct access to model's weights. Can the authors expand more on their motivation?
>
> What we mean is that our watermark is the first to have any provable robustness guarantee when the attacker has access to the model’s weights. While any watermark could be deployed in an open source setting, existing schemes would be trivially removable. In contrast, we prove that even when the attacker has this knowledge, removing our watermark requires degrading the quality of the model. This shows that watermarking indeed has a benefit in an open source setting. The benefit of watermarking when the attacker has direct access to the model’s weights is that it gives us the capability to identify model-generated content. The fact that the attacker has access to the weights makes it challenging to construct a watermark that is not easily removable; this is exactly the problem we study in this paper.
>
> > The proposed approach embeds watermark signals to the bias of the last layer's neurons. There is another approach by ByteDance that injects watermark into the LLM weights by finetuning (https://arxiv.org/pdf/2403.10553). Why is there no comparison with this approach? Infact, why is there no comparison with other watermarking schemes at all?
>
> The ByteDance paper achieves only heuristic robustness but not provable unremovability. We focus on provable robustness/unremovability guarantees, and are the first to achieve any unremovability guarantee in the open-source setting. We do not perform an experimental comparison, as our main contribution is not a scheme with optimized practical parameters. We will include more thorough discussion of related work, such as ByteDance.
>
> > There are adaptive ways to bypass watermarks. One is by using adaptive paraphrasers. If the proposed watermark scheme is unremovable, yet detectable, why are there no empirical results proving the 'unremovability' claim using adaptive paraphrasers, or even normal paraphrasers like Dipper, or even using open source LLMs for paraphrasing.
>
> Our unremovability guarantee is that an attacker without sufficient knowledge of high-quality text must either compromise quality or fail to remove the watermark. The amount that quality is compromised depends on the attacker’s knowledge of high-quality text. Using paraphrasers is a practical instantiation of this tradeoff. If the paraphrasers largely preserve text quality and remove the watermark, it is because they leverage knowledge about high-quality text. On the other hand, paraphrasers that do not embody sufficient knowledge about high-quality text will yield poor-quality text but may remove the watermark. We acknowledge that an attacker with access to a high-quality paraphraser, or unwatermarked open-source model, can remove the watermark– this is inevitable, and this is why we prove unremovability only against an attacker with limited knowledge.
>
> > How efficient is the detection process? How many tokens does it require to detect the proposed scheme, especially using its optimal hyperparameters? I feel the experiments the authors provided to prove the efficiency and strength of this approach are not enough.
>
> We show that our watermark is detectable in 300-token responses. We do not attempt to optimize our parameters, as we aim to show a proof-of-concept and not a deployment-ready scheme. Our primary message is that open-source watermarking is theoretically possible, and that our theoretical modeling assumptions are reasonable enough that our provable results carry over into practice. We leave concretely optimizing the scheme in practice to future work.

---

> ### Comment · Reviewer_p6Eu · 2024-11-25
>
> Thank you for your response.
>
> - If I understand correctly, your approach addresses threats in scenarios where an attacker has full white-box access to the model's weights. In this context, the threat model assumes that the attacker aims to remove the watermark from the model without compromising the quality of the text it generates. Your unremovability guarantees suggest that this is either impossible or highly unlikely under these conditions. While the model-based removability guarantees are theoretically interesting, I am concerned that this scenario may not align with practical applications, as most LLM providers typically only offer black-box access. I may be wrong, maybe you should try to explain your motivations better.
>
> - For open-source LLMs, an attacker with white-box access is unlikely to alter the model's parameters if high-quality paraphrasing can remove watermarks effectively. I mean, why would I try to remove the watermark from the model's param and risk degrading text quality if I have access to tools like GPT-4, Gemini, Claude, or other well-trained open-source LLMs that can easily generate high quality paraphrased versions of watermarked text. This suggests that unremovability should be inherent to the text itself, rather than dependent on the model, and that, intuitively is impossible.
>
> - To strengthen your claims,why not provide empirical evidence demonstrating that your theoretical guarantees hold practical value. A thorough evaluation against various attacks such as translation, paraphrasing, etc, would greatly pass the message across, only if you beat other watermarking algorithms.
>
> For now, I will maintain my current score but encourage you to address these points to make a stronger case for your contributions.

---

### Official Review · Reviewer_Pxpj · 2024-10-29

**Soundness:** 2
**Presentation:** 2
**Contribution:** 2
**Rating:** 3
**Confidence:** 4

**Summary:**

The paper proposes a method for embedding watermarks in large language models (LLMs). This method incorporates watermark information by adding noise that follows a normal distribution to the model's output, with the noise serving as the watermark's key. The authors also demonstrate that, under certain assumptions, the embedded watermark is unremovable. The feasibility of the proposed scheme is validated using the OPT-6.7B and OPT-1.3B models.

**Strengths:**

1. The authors attempt to embed noise information following a normal distribution as a watermark into the text output of the model. This is an interesting endeavor that could potentially aid future watermark embedding algorithms.

2.The paper attempts to theoretically discuss the unremovability of watermarks, which is also an interesting analysis.

**Weaknesses:**

1. The paper's description of the algorithm is not clear enough and does not reflect the specific implementation of the watermark embedding algorithm. There is watermark detection for text in Algorithm 4, but there is no embedding algorithm for text.

2. The paper discusses the unremovability of watermarks, which is generally referred to as robustness in other papers. The paper does not compare the robustness of its approach with those of other papers. It should also discuss the soundness property of the watermark, which typically contradicts robustness.

3. The writing in the paper is not clear enough, which makes it difficult to understand the algorithms it contains. Specific issues will be provided below.

**Questions:**

1. What is the relationship between Algorithm 4 and Algorithm 3? If Algorithm 4 is the primary method for text watermark detection, then when is Algorithm 3 invoked?

2. How is the symbol \Delta (x_i) defined in Algorithm 4? How is it calculated?

3. In watermark detection, each token should be evaluated. Why is it necessary to check x_i \in S in line 4 of Algorithm 4?

---

### Official Review · Reviewer_8Mtu · 2024-11-04

**Soundness:** 2
**Presentation:** 1
**Contribution:** 1
**Rating:** 3
**Confidence:** 5

**Summary:**

The paper focuses on the problem of LLM watermarking and proposes a scheme applicable to open-source models. The key claimed advantage of the scheme is its provable unremovability which the authors rigorously derive. Experimental results on two OPT models are shown, including the evaluation of robustness to token substitution and Gaussian perturbations to model weights.

**Strengths:**

- As the authors recognize, watermarking of open-source LLMs is one of the most important open problems in current generative model watermarking research, so studying it has the potential for high impact.

**Weaknesses:**

Unfortunately I believe the paper in its current state is far from being able to deliver that impact. Namely:
- While I agree that some definition must exist, formalizing "LLM that produces high-quality text" as "closeness to the original LLM in weights of the last bias layer" seems arbitrary and far from realistic notions of quality. This greatly simplifies the problem, unfortunately making the theoretical results (claimed key contribution) largely not relevant. While I appreciate the rigor and work authors have put in proving the results, formalizing the intuition that a random vector in N-dimensional space is unlikely to match a particular unknown direction, I unfortunately do not think this provides any valuable insight in terms of the robustness of an OSS watermark to realistic transformations.
- Given this, the blanket claims that the watermark is "unremovable" (title, abstract, introduction) render as dangerous overclaims that may cause confusion in the field if the paper is accepted. These should be greatly adjusted and qualified to explain the peculiar definition of quality. To actually get a meaningful notion of unremovability, the authors could consider realistic transformations commonly applied to OSS models such as finetuning, PEFT, quantization, pruning, or at least random modification of all weights (the argument on L129/130 is unclear). These are currently neither discussed in theory nor included in the evaluation. Interestingly, the authors recognize that prior work Gu et al. (2024) considers fine-tuning yet do not consider this themselves.
- As the authors also recognize, the proposed scheme is a variant of UnigramWatermark. While scheme simplicity is not a weakness per se, interesting/novel technical aspects of the proposed scheme are also not a strength of this paper. This is further harmed by the fact that popular LLMs often do not use the final layer bias, making the proposed scheme inapplicable. In fact, this is true for OPT models used in this work (https://github.com/huggingface/transformers/blob/v4.46.0/src/transformers/models/opt/modeling_opt.py#L1052), bringing into question the current evaluation.
- LLM watermarking, which this paper positions itself as part of, generally focuses on detecting LLM-generated outputs. Yet, this paper starts from the related but different notion of detecting that a model was based on a watermarked model from its weights, and prove key results in this case. This is a new scenario which is unexplained and unmotivated, should be explicitly separated from the common understanding of LLM watermarking promised in early parts of the paper, and raises many questions. For example, if we assume transformations of our OSS model change nothing but the final bias layer, can't we use the other (unchanged) weights to demonstrate that the resulting model was made from our model?
- Evaluation has many drawbacks, among else it does not include any baseline (such as Gu et al. (2024)), uses high FPRs, and uses no realistic attacks on text such as paraphrasing, generally used in prior work. As the authors note, the performance of the watermark is below non-OSS baselines, which is to be expected, but does not present a case for this method as useful beyond the OSS case.
- The paper is written and presented in a confusing and convoluted way, seems to be written in a rush, and it is often very hard to understand the key parts. I include some examples/suggestions below, in hopes that this helps the authors get insight into the issues and improve their writing in the future to the level expected at ICLR. I am happy to further assist the authors here if they have questions.
- (Minor) While this does not affect my assessment, L173 contains another dangerous claim, citing Zhang et al. (2024) to say that any LLM watermark is removable. This is a misunderstanding of the original paper which studies an idealized case where a random walk on the space of "equivalent" documents is possible while preserving quality, and the random walk is rapidly mixing. To avoid misinforming readers, this citation should be appropriately qualified.

Overall, while I do appreciate the authors tackling such a hard and important problem, I do not see the contribution of the paper at this point, and believe it thus to be clearly below the bar for acceptance.

---

The list of writing/formatting/presentation comments for the authors follows, which I hope they find helpful. I do not expect authors to reply to each point, although I should be corrected if I misinterpreted some points.
- L53: the phrase "unremovability from a modest amount of text" is confusing and should be made more precise
- L54-60 seems to repeat the same point about the adversary twice, requiring several readings
- I appreciate the inclusion of the Overview section; however, instead of previewing and summarizing the technical details, this section is often the only place where concepts are explained in terms of the concrete instantiation of interest (LLMs). E.g., 4.1. does not reflect on what unremovability means in the setting we consider, but only provides abstract definitions. This makes the paper hard to read and understand the actual instantiation.
- Another choice that contributes to this is the use of "content" to counterinuitively often mean "last layer bias vector" instead of "text". Similarly in Alg. 3 it is not made clear if "original content" refers to the watermarked or pre-watermarked model weights; effort by the reader is needed to understand this.
- Sec. 2 uses (M*, M') for (original, watermarked) model, inconsistent with ($w^\star, w_{wat}$) below, causing some confusion.
- L87: "checking the correlation" is quite imprecise
- L104: why the region is a halfspace is not explained; while it is a simple property of dot product, this should be made explicit to help readers grep this part
- L107: "add to $w^*$" is unclear. I suspect this should say "the adversary can add a vector to $w_{wat}$" instead; this should be made precise.
- L230: Q should probably be L? Such mistakes should be especially avoided in key definitions.
- L231: "p.p.t." should be defined, I believe it is not a standard term in this community
- L318: logits $l_i$ seem to refer to values before adding any bias? This is very ambiguous and should be made clear in the writing.
- "Quality score" shows up first in Fig. 3 but is not previously introduced which is quite confusing.
- The paper has no figures before the evaluation which is highly unusual, especially as there ary many instances where a visualization would greatly aid understanding (e.g., halfspaces / gaussians in the model parameter space). I suggest the authors generally consider this when writing.
- The margins on top of each page are very small which suggests the style file was tweaked. Note that the ICLR formatting instructions state "Tweaking the style files may be grounds for rejection.". While the paper doesn't seem to have been desk rejected in this instance, I strongly suggest the authors remedy this and follow the instructions.

**Questions:**

- In Theorem 1 shouldn't the distribution over $w^\star$ be centered at $w_{wat}$ and not 0? This is also present in the proof and Theorem 3. Is this a mistake or my misunderstanding of the statements?
- L145 states that Aaronson (2022), Christ (2024) and Fairoze (2023) are based on partitioning the tokens into red and green lists. Can you elaborate on this view of these methods, as my understanding was that they are quite different and do not use the red/green concept?
- Were OPT models modified to use the final layer bias to enable experimental evaluation?

---

> ### Author Response · Authors · 2024-11-25
>
> In answer to your questions:
> > In Theorem 1 shouldn't the distribution over $w^*$ be centered at $w_{wat}$ and not 0? This is also present in the proof and Theorem 3. Is this a mistake or my misunderstanding of the statements?
>
> You are correct; these distributions should be centered at $w_{wat}$.
>
> > L145 states that Aaronson (2022), Christ (2024) and Fairoze (2023) are based on partitioning the tokens into red and green lists. Can you elaborate on this view of these methods, as my understanding was that they are quite different and do not use the red/green concept?
>
> Aaronson and Christ et al. use hash functions evaluated on a previous portion of the response to determine a set of tokens whose probabilities to increase in the next sampling step. One can interpret this set of tokens with increased probabilities as the “green” list, and the other tokens as the “red” list. In this view, Aaronson and Christ et al. randomly choose a green/red partition in each sampling step, and increase the probabilities of the green tokens. Fairoze et al. is similar except that the green/red lists consist of n-grams rather than individual tokens. That is, Fairoze et al. randomly choose a set of (“green”) n-grams whose probabilities to increase in each sampling step.
>
> > Were OPT models modified to use the final layer bias to enable experimental evaluation?
>
> Yes, OPT models were modified to use the final layer bias. We take the unwatermarked model to be OPT with all zeros as the final layer biases, and report quality scores for unwatermarked text as such. The same could be done for a model that is released open-source. Ideally, the biases in the last layer should be trained rather than set to all zero as in OPT. The fact that we assume that the last-layer biases are trained does restrict the class of models we consider; however, this class is still quite broad, and if one wished to release a watermarked model one could easily train the last layer biases.

---

### Author Response · Authors · 2024-11-25

We thank the reviewers for their detailed and constructive comments, and respond here to weaknesses brought up in multiple reviews. We respond to individual questions below.

- Unremovability vs. robustness: We consider a far stronger attack setting than existing inference-time watermarks that change the sampling algorithm of the model (e.g., Kirchenbauer et al., Aaronson et al., Christ et al., etc). In our attack setting, the adversary has access to the weights and code for the model. We call resistance to this stronger attack “unremovability,” which is similar to robustness although against a stronger attacker. We emphasize that existing robust inference-time watermarks are easily removable–the attacker can easily modify the code for the sampling algorithm so that it does not embed the watermark at all. As unremovability is a new property, there are no existing works that prove (even a weak form of) unremovability. Therefore, we focus on theoretical, provable unremovability, showing surprisingly that it is possible at all. We reinforce our theoretical results with experiments, which should be viewed as a proof-of-concept rather than evidence that our watermark is optimal compared to inference-time watermarks. While not optimal, we do show standard robustness and quality evaluations used in inference-time watermarks, and our watermark has reasonably good robustness and quality (Figures 3 and 4).
- We emphasize that we show that it is possible to have open-source watermarks with provable unremovability and reasonable detection rates and robustness in practice, as shown in our experiments. We use similar experiments as in the literature; we use the same substitution attack robustness evaluation as that of Kuditipudi et al., and we use a similar quality score to that of Piet et al. We acknowledge that compared to inference-time watermarks, ours is not optimal; however, it performs reasonably well.
We especially thank the reviewers for the editorial suggestions and will improve the clarity of the writing.

---

### Meta-Review · Area_Chair_4iP7 · 2024-12-21

**Metareview:**

This submission describes a watermarking approach for open-source LLMs. The submission correctly identifies that a desirable property of a watermark embedded in an open-source model would be that it is not easily removable. The approach injects a uni-gram/zero-context watermark in the sense of Zhao et al. 2024, by modifying the biases of the last linear layer of the model. The authors then formalize a particular notion of unremovability under which they show that this watermark is unremovable.

Reviewers mainly disagree with applicability of the theoretical findings in this work. The way the authors formalize the notions of unremovable watermark and high-quality text, were found not acceptable. Many reviewers also note that baseline attacks against models with open-source watermarks should just be the same as attacks against watermarked API models, i.e. using adaptive attacks, or paraphrasing.

During the response, the authors retreat to the position that their proof is sound, but this is not in disagreement. The applicability of the proved unremovability, and statement of the assumptions are in question.

On a different note, from my reading, I would also point out that there are simple baseline attacks that should have been considered. The analysis works based on Gaussian modifications to the bias being hard, but an attack can easily 1) finetune the model to obtain a new bias layer (especially as current-generation models do not derive substantial quality from their last layer bias, if they include one at all), or 2), given that the watermark is based on a last-layer bias modification, it is input independent, and a simple tabulation of unigram probabilities over sufficient quantities of text will approximate the exact bias vector that needs to be removed.

Finally, I am not convinced that Assumption 3 holds in the setting proposed in this work, where the watermark is input-independent by design.


I do not recommend acceptance of this submission. I do think the review period was productive, and I hope the authors incorporate the received feedback into future versions of their material.

**Additional Comments On Reviewer Discussion:**

This metareview is based on the review of 8Mtu and my own understanding of the submission. I also include interesting comments from p6Eu and p4oF.

---

### Decision · Program_Chairs · 2025-01-22

Reject